# Health Locus of Control in Parents of Children with Leukemia and Associations with Their Life Perceptions and Depression Symptomatology

**DOI:** 10.3390/children7050040

**Published:** 2020-05-01

**Authors:** Marta Tremolada, Livia Taverna, Sabrina Bonichini, Maria Caterina Putti, Marta Pillon, Alessandra Biffi

**Affiliations:** 1Department of Development and Social Psychology, University of Padua, 35131 Padua, Italy; s.bonichini@unipd.it; 2Pediatric Hematology, Oncology and Stem Cell Transplant Center, Department of Woman’s and Child’s Health, University of Padua, 35128 Padua, Italy; mariacaterina.putti@unipd.it (M.C.P.); marta.pillon@unipd.it (M.P.); alessandra.biffi@unipd.it (A.B.); 3Faculty of Education, Free University of Bolzano-Bozen, 39042 Brixen, Italy; livia.taverna@unibz.it

**Keywords:** children with leukemia, parents, health locus of control, depression, life perceptions

## Abstract

In childhood cancer, parents have an important role in the promotion of their children’s wellbeing and in their adoption of a locus of control style towards their children’s health. The current study aimed at identifying types of locus of control in parents of children with leukemia and the possible association with depressive symptomatology and current life perception. One hundred and four parents were recruited at the Hematology–Oncology Clinic of the Department of Woman’s and Child’s Health, University of Padua, one month after a leukemia diagnosis. Participants were Caucasian with a mean age of 37.28 years (SD = 5.89), mostly mothers (87.5%) and with a mean of 12.16 years of education (SD = 3.82). After signing the informed consent, they filled in the Ladder of Life, the Brief Symptom Inventory-18 and the Parental Health Locus of Control (PHLOC) questionnaires. Paired-samples *t*-test (*t* = −14.42; df = 103; *p* = 0.0001) showed that parents of children with leukemia were more inclined to have an external locus of control than an internal one. The hierarchical regression analysis model (R^2^ = 0.34; F = 4.32; *p* = 0.0001) identified health professional influence (ß = −0.28; *p* = 0.004), current life perception (ß = −0.3; *p* = 0.013) and future life perception (ß = −0.26; *p* = 0.012) as significant predictors of parental depression. Current life perception was best predicted (R^2^ = 0.25; F = 3.96; *p* = 0.01) by the parental influence locus of control style (ß = 0.25; *p* = 0.03). Improving trust in the medical staff care and strengthening the internal locus of control in parents could be a preventive program to cope with parental depression symptomatology.

## 1. Introduction

Health locus of control (HLOC) is defined as the set of beliefs a person has about his or her personal influence on health. This set of beliefs includes an internal locus of control (if the individual believes that personal actions or thoughts can affect their outcomes) and an external locus of control (if the outcome is believed to be determined by powerful others such as God or health professionals, or if chance is believed to control the outcome) [1,2]. Empirical research suggests that the health locus of control may play a significant role in determining people’s health-related behaviors [3] and can explain some of the variability in health-seeking behaviors or attitudes [4]. In childhood cancer, parents have an important role in the promotion of their children’s health especially when their children are very young, with parenting attitude considered to be a key factor in helping children cope with their illness [5]. The parental beliefs towards their children’s health could have a role in determining their educational and supportive actions towards ill children and for this reason it is fundamental to identify the stable factors that could help parents to have a good life perception and to avoid depression symptoms. It is therefore of interest to assess the parental locus of control relative to their children’s health [6].

Attribution research has examined parental health locus of control (PHLOC) as an important component of parental cognitions and beliefs [7]. PHLOC is the parental perceptions of their power and efficacy in the parent–child relationship [8]. Although it has often been conceptualized as being either internal or external (e.g., [9]), some researchers have suggested that it is a continuous construct (particularly as it relates to motivation) and ranges from internal at one end to external at the other [10]. Parents with a more internal PHLOC attribute their child’s behavior to internal factors, such as their own parenting techniques and strategies, whereas parents with a more external PHLOC attribute their child’s behavior to external factors, such as chance or the negative influence of peers, rather than their own parenting behaviors [9].

Parents who reported a strong belief in the influence of external powerful others, such as medical staff, had children who were more successful throughout treatment. In contrast, those parents who reported that their child’s outcome was due to fate or to mass media had children with worse treatment outcomes. Regarding adherence, parents with strong beliefs in fate and those who felt they were more responsible for the child’s weight problem (i.e., higher internal PHLOC) attended fewer therapeutic sessions [11]. This study suggests that parents who place confidence in powerful others such as treatment providers have children who are more successful in treatment. Perhaps more confidence in powerful others indicates a greater willingness on the part of parents to accept and implement professional recommendations that a structured treatment program provides. Parents who rated themselves as more responsible for the child’s weight problem had children with marginally less successful outcomes in treatment. Further, parents who rated themselves as more responsible for the child’s weight problem had children with significantly poorer attendance.

The association between belief in powerful other people and treatment outcome is significant, given the prominent role that professionals play in facilitating treatment.

### 1.1. Life Perception and Depressive Symptomatology in Parents of Children with Cancer

In general, the literature converges upon two major facts. The first is that the most difficult period for parents and families happens just after the diagnosis [12], when the child undergoes several invasive medical procedures (e.g., bone marrow aspirates, lumbar punctures) and treatments (e.g., chemotherapy). Several studies showed that the acute phase is the most stressful period for parents [13], during which a new family “reality” must be built up [14]. 

The second point highlighted by the literature is that, unfortunately, a notable percentage of parents are not able to make a pathway for good adjustment and quality of life and remain indelibly scorched by the experience: increased negative emotions such as anxiety, depression, insomnia or somatic and social dysfunction shortly after diagnosis have been found [15]. The incidence of post-traumatic stress symptoms (PTSS) is high in the first month of therapy [16], and generally throughout the first 6 months of therapy, and can be present for a long time [17]. Social support received by mothers helps them to have a good perception of their lives, is associated with less psychological symptoms and has been found to be predictive of PTSS [18]. Depressive and post-traumatic stress are different types of symptomatology that can be seen in parents of children with leukemia. These types of disorders could both coexist in parents, even if they represent different constructs in psychological health. Current life perception could be considered as a sort of short version of the perceived quality of life and future life perception could be comparable as a degree of hope. These scales could be a valid tool to screen people more at risk from depression symptomatology as precedent studies showed us [18]. For this reason, it is important to add this construct as a factor associated with depression. 

Another important predictor of depression can be the health locus of control. The literature on this topic did not focus on PHLOC, but on HLOC. However, HLOC studies could be indicative to understand better the possible associations between depression and health locus of control style in the individuals. Some people believe that their health status is controlled by themselves, they believe that they stay or become healthy as a result of their own behavior (called internal health locus of control). Others believe that their health status is controlled by powerful others or chance, so factors that determine their health are ones over which they have little control (called external and chance health locus of control, respectively) [19]. Perceived control might decline after a cancer diagnosis and during the process of aging [20]. Still, the relationship between internal control and psychological adjustment to cancer remains largely unknown [21]. What we do know is that to a large extent, perception of control can be taught. Overall, the internal health locus of control in older patients with cancer was associated with higher risk of depression, while the external “powerful others” locus with a lower one [22]. A significant relationship between perceived threat and depression was found only among participants reporting low levels of internal locus of control [22]. These findings support the hypothesis that perception of cancer as life threatening is an important factor in determining the level of depression among cancer patients. Moreover, the differentiation between internal and external health locus of control (HLC) suggests that internal HLC may be more relevant than external HLC in managing perceived threat. An internal locus of control can be interpreted as having a sense of agency and mastery which is important in managing the cognitive perception of the threat of illness [23].

### 1.2. Research Questions

We formulated the following research questions:(1)What are the reporting scores of parental life perceptions and depressive symptomatology after 1 month of therapy?(2)In this acute first phase we expected to find low scores in parents’ current life perceptions (with lower indexes as indicative of worse life perception), based on the above mentioned literature as it appears to be the most difficult period for parents (i.e., [15]). In addition, we expected a high incidence of parental depressive symptomatology, which we saw as a possible associated factor impacting on PTSS [18].(3)Which are the main parental health locus of control beliefs in relation to the child’s disease factors after 1 month from the diagnosis communication? Do parents show a more internal or external approach?(4)There are no specific studies on PHLOC in the context of pediatric cancer. Based on the literature on health locus of control (HLOC) in general, we could expect that parents would show both internal (such as their parenting role and in their psychological resources) [9], and external HLOC (such as health professionals or God) [11]. We hypothesized that parents’ internal adoption would be related to a positive perception of their life and that parents consider children’s influence (a typology of internal locus of control style) more important as the age of the child increases.(5)Could the type of PHLOC be associated with the depressive symptomatology and the current life perceptions in parents? What are the possible associations with demographic and illness factors?(6)Which are the best child and parent factors that could influence parental life perception and depressive symptomatology?

The negative emotional life perceptions could impact on depressive symptomatology. Dealing with the health locus of control, we do not have a clear idea which type of locus of control (internal vs. external) may dampen depressive symptomatology or negative life perceptions. We also expected that negative life perceptions and depressive symptoms could be worse in parents whose children are affected by high risk leukemia, compared to standard risk, based on the different prognosis indicated to the parents. 

## 2. Materials and Methods

Participants included 104 parents of children with leukemia after the first month from the diagnosis communication recruited at the Hematology–Oncology Clinic of the Department of Woman’s and Child’s Health, University of Padua. All parents were Caucasian with a mean age of 37.28 years (SD = 5.89; range: 19–58), mostly mothers (87.5%) and with a mean education expressed in years of 12.16 (SD = 3.82; range: 5–20). Parents’ incomes were mostly average (55.3%), followed by high (23.3%) and low (21.4%), but above poverty. The average of job hours/weekly was mostly around 35 (28%) and 45 (22%). Some parents were temporarily relieved of their work or were housewives (44.7%). The parents who participated were mostly mothers (N = 91) and only a few were fathers (N = 13), because the mothers were more proximal to the child during hospitalization while fathers stayed with other siblings or continued to work to maintain the family. In the preliminary analysis we controlled the possible differences between fathers and mothers. There were no significant differences in all the variables considered, so we decided to consider them all together. 

Children’s average age was 5.94 years (SD = 4.12, range = 1–17 years), there were 50 females and 54 males. The majority of children were affected by Acute Lymphoblastic Leukemia (ALL) (N = 87), while 17 had Acute Myeloid Leukemia (AML). Considering the risk band related to survival percentage, the patients belonged to the standard risk (N = 19), medium risk (N = 69) and high risk (N = 27) bands. 

### 2.1. Procedure

The present study was part of a major research project entitled: “Family factors predicting the short- and long-term adaptation and quality of life in children with leukaemia” approved by the psychological research Ethical Committee, protocol number 2313. The parents were contacted by a clinical psychologist during the first hospitalization of the children. The project aims were explained, and informed consent was asked for. Informal contacts with the participants were kept up on a daily basis to provide support and motivation for the project. The participants were informed that they were free to drop out at any moment of the study. Socio-demographic information, the Ladder of life questionnaire, BSI-18 (Brief Symptom Inventory-18) and Parental Health Locus of Control (PHLOC) questionnaires were administered to parents 1 month after the diagnosis.

### 2.2. Instruments

#### 2.2.1. Ladder of Life (CCSS) 

The parent had to evaluate, using a 1 (worst) to 10 (best) points scale, the quality of her/his present life, the quality of her/his life 5 years before the child’s disease and how satisfying her/his life would be in the future (5 years later from their son’s/daughter’s diagnosis). With this instrument, we could obtain information about individual perception of the past, the present and the future (low scores = worse perceptions). It has been administered to 118 Italian mothers of children with cancer, demonstrating good global internal consistency (Cronbach’s alpha = 0.73) [17].

#### 2.2.2. PHLOC [2]

This is a 30-item questionnaire assessing parent’s type of internal or external locus of control with respect to their child’s health. The PHLOC was used to assess parents’ beliefs about the health of his/her child. Child’s wellbeing can depend on destiny (absolutely not controllable), on external information sources (pediatric staff), or on the parent (fully controllable). The questionnaire assesses beliefs on the child, divine, fate, media, parental, and professional influences over the child’s health. For example, the fate subscale provides an index of the extent to which parents believe that the health status of their child is predominantly a matter of luck (e.g., whether my child avoids injury is mostly a matter of luck). The American standardization (2) showed internal consistency reliability coefficients above 0.70 for all scales and test-retest (r) correlations all above 0.60, confirmed by the Italian standardization, which showed good internal consistency of the sub-scales (r > 0.70), and adequate test–retest correlations (r > 0.80) [24]. 

#### 2.2.3. Socio-Demographic and Medical Data

Each parent filled in a socio-demographic questionnaire with inquiries into their highest year of schooling, their education, their perceived economic situation, their type of home situation, their romantic relationship and their type of employment.

#### 2.2.4. BSI-18 [25]

The Brief Symptom Inventory 18 (BSI-18) consists of 18 items grouped into three dimensions of six items, serving as a screening for depression, somatization and anxiety. Respondents were asked to refer about how they felt over the last 7 days, and each item was rated on a 5-point Likert scale from 0 (not at all) to 4 (extremely). BSI-18 was used to assess psychological outcomes in parents of children under treatment for leukemia [18], demonstrating good internal consistency for both the Global score (Cronbach’s alpha = 0.92) and the specific domains (Depression: alpha = 0.84; Somatization: alpha = 0.83; Anxiety: alpha = 0.83). This a measure of both psychological mood symptoms and stress intensity that assesses the acute presence of these type of symptoms in parents and it could be indicative of their psychological health.

## 3. Results

### 3.1. Parent’s Current, Future Life Perceptions and Depressive Symptomatology after 1 Month

What are the reported scores of parental life perceptions and depressive symptomatology after 1 month of therapy?

To answer this question, we ran descriptive statistics. Current life perception was very low, while future life perception was instead reported at high levels. Depressive symptomatology in parents attested for 22% at high levels from scores of 3 (moderately) to 4 (extremely) scores and for the majority (78%) at none (0, not at all) or low levels (scores of 1 or 2). The depressive symptoms were the most reported, followed by respectively the anxiety and somatization scales (Table 1). 

### 3.2. Parental Health Locus of Control: Typology

What are the main parental health locus of control beliefs in relation to the child’s disease factors after 1 month from the diagnosis communication? Have parents shown a more internal or external approach?

To answer the first question, descriptive statistics were run on the several types of parental locus of control (Table 2). We can note that parental influence, an internal locus of control, was the most used, even if also medical staff influence and divine influence, external ones, were also frequently reported.

For the second question, once we had divided the health locus of control of parents into internal (parental and child influence) and external (divine, fate, medical staff and media influence) categories, we wanted to verify if parents adopted more internal or external styles towards the health of their children. Paired-samples *t*-test (*t* = ‒14.42; df = 103; *p* = 0.0001) within this group of parents showed that the external PHLOC scales (mean = 7.48; SD = 1.52) were more adopted than the internal ones (mean = 10.81; SD = 2.55). 

### 3.3. Pearson’s Correlations between Parental Health Locus of Control, Life Perceptions, Depressive Symptomatology, Demographic and Illness Factors

Could the type of PHLOC influence the depressive symptomatology and life perceptions in parents? 

We first ran Pearson’s correlations between the several interested variables. Table 3 shows these correlations underling significant associations with the above-mentioned variables. 

Child’s type of leukemia (r = −0.302; *p* = 0.02), child’s age (r = 0.23; *p* = 0.017) and parent’s age (r = 0.211; *p* = 0.02) were significantly associated with parental current life perceptions; younger parents of children with AML and with a younger age judged their lives as worse.

A child’s age was positively associated with the child’s influence on health control. The findings revealed a significant association within the scale of parental health locus of control between parental influence, medical staff influence and child influence. Fate and divine influence were instead related to each other. 

Depressive symptomatology in parents was significantly associated with current life perception (r = −0.51; *p* = 0.0001) and negatively with future life perceptions (r = 0.46; *p* = 0.0001). Life perceptions and depressive constructs were significantly associated because they both assessed an important piece of reported quality of life: telling how parents perceive their lives could be a screening tool to assess the possible depressive symptoms. Health locus of control of parental influence (r = −0.21; *p* = 0.02) and medical staff influence (r = −0.35; *p* = 0.0001) were associated with parental depressive symptomatology. 

Current life perception at the diagnosis was significantly associated with all of the three scales of parental health locus of control cited above.

Moreover, significant correlations were found between the internal locus of control and the following variables: child’s age (r = 0.20; *p* = 0.03), parental current life perception (r = 0.40; *p* = 0.001), parental depressive symptomatology (r = −0.23; *p* = 0.01). External locus was correlated moderately only with parental current of life perception (r = 0.20; *p* = 0.04). 

### 3.4. Significant Predictors of Parental Current Life Perception and Depressive Symptomatology

Which are the best child and parent factors that could influence parental life perception and depressive symptomatology? 

We first ran an ANOVA analysis with Bonferroni post-hoc correction to see if the leukemia’s risk band influences the parental ladder of life and depressive symptomatology. A significant mean difference was identified for parental current life perception (F = 3.53; df = 2; *p* = 0.03), with parents of children with a high risk leukemia showing a significantly lower reported score than those belonging to a standard risk one (Mean difference = −1.60; *p* = 0.04). The same trend was obtained for parental depression symptomatology (F = 3.16; df = 2; *p* = 0.04), with parents of children with high risk leukemia reporting higher indexes than those of children with low risk (Mean difference = 0.70; *p* = 0.04).

Secondly, we ran hierarchical regression analysis entering the child (age, type of leukemia) in the first step, parents’ socio-demographic factors (age, gender, years of schooling) in the second step, and parental health locus of control typology in the third one to identify the best predictors of parental current life perception. The first model was the best one (R^2^ = 0.13; F2 = 8.34; *p* = 0.0001), with child age (ß = 0.28; *p* = 0.003) and child diagnosis (ß = −0.26; *p* = 0.007) impacting significantly on parental current life perception (Table 4). Furthermore, the third model explained more of the proportion of variance with only parental influence as a predictor (ß = 0.25; *p* = 0.03).

Then, another regression analysis measured the possible factors associated with parental depressive symptomatology, entering the child (age, type of leukemia) and parent’s socio-demographic factors (age, gender, years of schooling) in the first step, parental current and future life perceptions in the second step, and parental health locus of control typology in the third one.

The third model explained the highest proportion of variance (R^2^ = 0.34; *p* = 0.04) with parental current life perception (β = −0.28; *p* = 0.015), future life perception (β = −0.28; *p* = 0.005) and medical staff influence (β = −0.28; *p* = 0.004) impacting upon parental depressive symptomatology (Table 5).

## 4. Discussion

There is extensive literature on the psycho-social consequences of child cancer on parental well-being and quality of life. The literature shows that mothers with older AML children hospitalized for more days, with less education, with more stressful life events and with more cognitive problems in the first weeks after the diagnosis are at a major risk of post-traumatic stress symptoms [17]. The most difficult period for parents happens just after the diagnosis [13] when the child undergoes several invasive medical procedures (e.g., bone marrow aspirates, lumbar punctures) and treatments (e.g., chemotherapy), and when a new family “reality” must be built up [14].

Unfortunately, a notable percentage of parents are not able to make a pathway for good adjustment and quality of life and remain indelibly scorched by the experience [26], showing anxiety or depression levels higher against normative data [15]. For this purpose, parental life perceptions could be a valid instrument to easily screen the parental psychological health as we have showed in a previous study [18]. For these reasons we wanted to know how parents perceive life in relation to the child’s disease and demographic factors after 1 month from the diagnosis communication. 

In this acute first phase we expected to find lower scores in parents’ current life perceptions and a high intensity of depressive symptomatology, basing on the existing literature. We also hypothesized that some factors could impact on the parents’ perceived life, like the type of diagnosis (with parents of children with AML having more troubles than those of children with ALL), the child’s age (parents of older children being much more worried) [17] and parents’ age (with a better situation with increasing age).

Our findings partially confirmed these hypotheses: a child’s diagnosis and age significantly influenced parents’ current life perceptions, but not parents’ age. Parents of children with AML perceive their current life to be worse than parents of children with ALL. This finding is also sustained by the literature [17,27]. In addition, the high-risk band negatively influenced parental current life perception and increased depressive symptomatology. The risk band could be more psychologically dangerous for parental mental health and also put the children’s physical health more at risk, with more intense therapy intensity and sequelae from therapy. As far as a child’s age is concerned, we found that older age is a negative factor causing a worse negative parental current life perception. At this purpose, we can argue that the increasing age of the child is positively associated with more psychological difficulties [28], with the consequence of causing crisis for parents in their caregiving role, and increasing the possible negative psychological outcomes. 

There have been no specific studies on parental health locus of control in the context of children with cancer. Parental health locus of control has been considered as internal if the direct principal agents of parent and child could potentially influence the child’s health, while it has been defined as external if other agents such as T.V., media, fate and health professionals had a greater influence [2]. Based upon the literature on health locus of control, we expected that parents invested a lot in both internal and external locus of control styles. We found that parents surprisingly adopted more external locus of control styles than internal ones. Childhood cancer probably put parents in an impotent state due to the extreme illness uncertainty and the need to rely on medical staff, media, T.V., and internet sources that give them a sort of security for their children’s health.

As a result, we studied the associations between parents’ locus of control and their own life perceptions and the child’s disease and demographic factors after one month from the diagnosis communication. Our findings showed that parental influence (a type of internal locus of control) is the most reported, followed by these external health control dimensions: medical staff influence, divine influence and child influence. Then we found that a parent’s current life perception was predicted by parental influence one-month post diagnosis. Parents probably think that they can act to ameliorate their life satisfaction probably due to their emotional coping strategies in the first period of hospitalization.

On the other side, another finding showed that parental depressive symptomatology was influenced by parents’ current and future life perceptions and their medical staff influence locus of control style. Current life perception gives us a measure to identify parents who are more in difficulty in their parental role and self-esteem about caring for the child during the illness. Positive future life perception can be considered such a measure of hope that appears to be a protective factor in developing depressive symptomatology. Medical staff influence is confirmed as an important health locus of control style that helps parents to dampen their possible depressive status. 

This information should be taken into consideration by health professionals to understand the degree of therapy compliance and to implement their perceptions about having an active role in controlling their child’s health. Communication strategies adopted by health professional in their relationship with parents could help in all treatment phases [29]. From a recent study on adult cancer patients [23], we know that an internal health locus of control might be more relevant than an external one in managing perceived threat and the psychological functioning, and so it is fundamental to also implement this also in parents of children with leukemia. This is confirmed in our study for positive life perceptions, but not for depressive symptomatology where the protective factor derives from the external “powerful others” locus [22], specifically the medical staff influence.

This study shows some limits. One is the different distribution of mothers and fathers in our participants. Mothers outnumber fathers, and so these results could be more representative for the mother’s experience than the father’s one. Another limit is that possible differences in parental locus of control, depressive symptoms and life perceptions with different family structures were not assessed (i.e., single parent, extended family involvement, LGBTQ parents). 

The strengths are that important clinical ramifications could be taken from this study, such as whether parental HLOC can be altered using behavioral or psychological interventions to more positively impact children during their diagnosis and treatment. This topic has not yet been explored in the literature and this study could indicate a new direction for clinicians and researchers.

## 5. Conclusions

Parents reported low indices of current life perception in the first month of therapy, although the depressive symptomatology had a higher intensity in only 22% of them. Parents of older children with AML or with more high-risk therapy that have a minor adoption of parental influence health locus of control had a lower current life perception.

The type of parents’ health locus of control can be a stable factor that influences depressive symptomatology, specifically the perceived media influence, together with current and future life perceptions of parents. These empirical data could give some suggestions for health professionals in regarding their psychological support in two directions. The first one consists of improving parental trust in the staff and medical care. This could be achieved with more efficacious communicative strategies trained by specialist health professionals. The second one deals with dampening parental impotence so to increase their self-esteem in helping children to improve their daily health. These results could be obtained with more supportive psychological interventions focused on parental health.

## Figures and Tables

**Table 1 children-07-00040-t001:** Descriptive statistics of life perceptions and of depressive symptomatology.

	Range	Mean	SD
Current life perceptions	1–10	4.43	2.21
Future life perceptions	1–10	8.03	2.03
Depression symptoms	0–4	2.29	0.99
Anxiety symptoms	0–4	2.16	0.90
Somatization symptoms	0–4	1.66	0.76
Global BSI score	0–4	2.03	0.77

SD = Standard deviation.

**Table 2 children-07-00040-t002:** Descriptive statistics of parental locus of control styles on child’s illness.

Locus of Control Styles	Range	Mean	SD
Parental influence (internal)	1–6	4.35	0.79
Medical staff influence (external)	1–6	4.30	0.83
Divine influence (external)	1–6	4.03	1.64
Child influence (internal)	1–6	3.13	0.97
Fate influence (external)	1–6	2.97	1.21
Media influence (external)	1–6	2.56	1.27

**Table 3 children-07-00040-t003:** Pearson’s correlations between Parental Health Locus of Control, Life Perceptions, Depressive symptomatology, Socio-Demographic and Illness Factors.

	Parental Influence	Fate Influence	Divine Influence	Child Influence	Medical Staff Influence	Media Influence	Parental Current Life Perception	Parental Future Life Perception	Parental BSI Depression
**Child’s current Age**	−0.047 0.635	0.053 0.595	0.212 * 0.032	0.360 ** 0.0001	0.063 0.524	−0.108 0.275	0.233 * 0.017	0.038 0.706	−0.20 * 0.02
**Child’s type of leukemia**	−0.190 0.053	−0.138 0.161	0.094 0.347	−0.053 0.590	−0.060 0.543	0.076 0.443	−0.302 ** 0.002	−0.123 0.215	0.07 0.42
**Parent’s Current Age**	0.011 0.913	0.036 0.719	−0.014 0.889	0.079 0.425	−0.043 0.667	−0.003 0.974	0.211 * 0.002	−0.123 0.215	−0.20 * 0.02
**Parent’s gender**	−0.187 0.058	−0.052 0.597	0.056 0.574	−0.040 0.688	−0.046 0.643	−0.203 * 0.039	−0.169 0.087	−0.052 0.598	0.01 0.86
**Parent’s schooling years**	−0.098 0.327	0.039 0.698	−0.054 0.589	−0.074 0.547	−0.201 * 0.041	0.155 0.119	−0.195 * 0.049	−0.135 0.175	0.08 0.36
**Parental influence**		0.042 0.676	0.049 0.626	0.477 ** 0.0001	0.379 ** 0.0001	−0.320 ** 0.001	0.373 ** 0.0001	0.152 0.125	−0.21 * 0.02
**Fate influence**			0.289 ** 0.003	0.185 0.059	0.152 0.124	0.098 0.324	0.152 0.122	0.088 0.379	0.04 0.62
**Divine influence**				0.164 0.100	0.172 0.083	0.090 0.368	0.149 0.135	0.108 0.281	−0.01 0.88
**Child influence**					0.316 ** 0.001	0.332 * 0.001	0.327 ** 0.001	0.095 0.338	−0.19 0.05
**Medical staff influence**						0.167 0.091	0.277 ** 0.004	0.189 0.056	−0.35 ** 0.0001
**Media influence**							−0.001 0.991	−0.009 0.932	−0.04 0.68
**Parental Current life perception**								0.54 ** 0.0001	−0.51 ** 0.0001
**Parental Future Life perception**									−0.46 ** 0.0001

BSI = Brief Symptom Inventory; * *p* value < 0.05 ** *p* value < 0.01.

**Table 4 children-07-00040-t004:** Hierarchical regression models predicting parental current life perception.

Step	Variables	R^2^	ΔR^2^	F	*p*	β	*p*
1	Child’s demographic and illness factors	0.13	0.15	8.34	0.0001		
	*Child diagnosis*					0.28	0.003
	*Child age*					−0.25	0.007
2	Parental socio-demographic factors	0.15	0.05	4.59	0.13		
3	Parental health Locus of control strategy	0.25	0.13	3.96	0.01		
	*Parental influence*					0.25	0.03

**Table 5 children-07-00040-t005:** Hierarchical regression predicting Parental depressive symptomatology.

Step	Variables	R^2^	ΔR^2^	F	*p*	β	*p*
1	Child’s and parent’s socio-demographic factors	0.02	0.07	1.78	ns		
2	Parental life perceptions	0.28	0.26	6.45	0.0001		
	*Current life perception*					−0.31	0.004
	*Future life perception*					−0.30	0.004
3	Parental health Locus of control strategy	0.34	0.09	4.82	0.04		
	*Current life perception*					−0.28	0.015
	*Future life perception*					−0.28	0.005
	*Medical staff influence*					−0.28	0.004

ns = not significant.

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
