# Peer review of "Health Locus of Control in Parents of Children with Leukemia and Associations with Their Life Perceptions and Depression Symptomatology"

_children, 2020, doi:10.3390/children7050040_

Round 1

Reviewer 1 Report

The paper is well designed with interesting finding regarding types of locus of control in parents of children with leukemia. These information can provide health-care providers with information to provide psychological support to parents of children with leukemia. I have some suggestions below for the author

  1. Please change "leukemia children" in the text to "children with leukemia". Leukemia is not what identify these children characteristics but rather just the disease that they have. 
  2. Why the author did not look at the disease prognosis to see if this has any effect on parental depression or life perception in table 2? I understood that the author indirectly looked at this through the type of leukemia variable, but prognostic factors like expected 5 year survival would directly answer the question of how much impact the expected disease outcome would influence parent's state of mind. 

Author Response

Comments and Suggestions for Authors

The paper is well designed with interesting finding regarding types of locus of control in parents of children with leukemia. These information can provide health-care providers with information to provide psychological support to parents of children with leukemia. I have some suggestions below for the author

Please change "leukemia children" in the text to "children with leukemia". Leukemia is not what identify these children characteristics but rather just the disease that they have.

We corrected it throughout the text. Thank you.

Why the author did not look at the disease prognosis to see if this has any effect on parental depression or life perception in table 2? I understood that the author indirectly looked at this through the type of leukemia variable, but prognostic factors like expected 5 year survival would directly answer the question of how much impact the expected disease outcome would influence parent's state of mind.

We run new analyses. See lines 348-354.

Reviewer 2 Report

Overall, the authors present research that is interesting for this patient population. The idea that parental LOC impacts parental depressive symptomatology, which can ultimately impact children is exciting to consider. There are also important clinical ramifications of this study, such as whether parental LOC can be altered using behavioral or psychological interventions to more positively impact children during their diagnosis and treatment. However, there are numerous grammatical errors that make the authors' argument difficult to follow. Additionally, correlation does not equal causation, and thus, results may need to be tempered.

Overall Strengths

  • Presents data about parental LOC, which there is limited research
  • Depression and LOC have extensive literature, but how it relates within the cancer population is limited, so this study fills in a needed gap
  • There are likely clinical applications that can be extended from the current study, such as interventions geared towards shaping positive parental LOC

General Weakness

  • The 3 hypotheses are difficult to understand, partly due to grammatical errors and inconsistencies, as well as overall research design.
  • Multiple grammatical errors throughout the paper that make is difficult to read
  • Inconsistent use of acronyms and terminology. For example, PLOC is introduced in line 44, but sporadically utilized. Health-related LOC and parental LOC and external/internal LOC are used interchangeably, which make it difficult to follow.
  • Inconsistent capitalization (e.g., Children vs. children)
  • Inconsistent spelling (e.g., haematology vs. hematology)
  • Depressive symptomology and PTSS are used almost as predictive of one another; however, these may be different psychological constructs
  • Correlations regarding parental LOC and depressive symptomatology may not be an accurate method of answering the research questions
  • Base rates for AML and ALL are significantly different, which is reflected in this study sample. However, given the different numbers of subjects in each cell, authors may not be able to accurately compare the 2 groups
  • The statistical model does not control for various factors, such as treatment intensity, length of hospital stay, etc.
  •  

 Introduction

  • 42-43: the assumption is that parental locus of control impacts parenting behaviors; however, the authors could have expanded this idea further as it seems rather abrupt. For example, does parental LOC impact parenting behaviors? And if so, how does it change parental behaviors?
  • 53-57: confusing argument, difficult to follow as the authors use examples of internal and external LOC. Furthermore, uncertain as to what authors are attempting to lay out, that external and internal LOC occur interchangeably and have both negative/positive outcomes? Or that a particular external LOC, specifically trust in other medical professionals is the only beneficial parental LOC related to treatment outcomes
  • 63-66: Reference needed; however, authors write about internal LOC having poor outcomes for children’s weight problems. This argument seems misplaced and may be better served in the previous paragraph.
  • 73-80: it may be beneficial to expand on what is PTSS. The end of this paragraph and the start of the next appear to make it seem that the authors are utilizing PTSS and depressive symptoms as synonymous

Hypotheses

  • Authors first introduce the concept of Current Life Perceptions, but this seems sudden; authors may need to further define this concept and how it relates to depressive symptomatology
  • 102: low scores in parents’ Current Life Perceptions, is this low score good or bad?
  • 104-105: high incidence of depression could be “predictive” for PTSS; however, the correlational design may not be able to answer this hypothesis accurately. (Correlation does not equal causation).
  • 109-114: 2nd hypothesis is that parents will show higher scores on both external and internal LOC? I’m assuming that the authors are trying to present hypotheses based on the subdomains of parental LOC in their questionnaire; however, they do not explain or provide information about the different domains of LOC in the introduction
  • 115-122: very difficult to understand the 3rd hypothesis

Methods

  • 126: Name of clinic changes compared to line 18
  • 130: reference to support authors SES stratification by “Italian norms”
  • 133-136: authors controlled for potential differences in mother and father responses. However, mother outnumber father (91 to 13), what statistical methods were utilized to control for these differences? The authors also do not present other psychosocial information, such as family structure (i.e., single parent, extended family involvement, LGBTQ parents)
  • 173: BSI-18 screens for depression, somatization, and anxiety, but given the acute nature of the situation (~1 month of cancer diagnosis), what is this measure capturing? Disease distress, psychosocial stress, or psychological mood symptoms?

Results

  • 189-190: what do the numbers 0.22% and 0.88% refer to? Additionally, these numbers do not add up to 1
  • 183 -190: information can be better presented in a table
  • 191: first time authors introduce the terms “internal or external adoption” as it relates to locus of control. Please provide background
  • 198-201: difficult to comprehend what authors are presenting. What do the authors mean when they write that they “divided” parents into internal and external categories? And then they ran a paired-samples t test to compare responses of which group?
  • 204-206: hypothesis 3 changes and lists that the authors will examine SES factors, which they had not presented before in the intro, at the 2nd month, which previously did not have a time frame
  • High correlations between parental current and future life perceptions with BSI were noted. How did the authors address the high correlation between these variables in their regression model?

Discussion

  • Repeats sentences found in the abstract and intro
  • Verbiage becomes too colloquial (i.e., line 247 “huge”)
  • No discussion of limit to the study

Author Response

Comments and Suggestions for Authors

Overall, the authors present research that is interesting for this patient population. The idea that parental LOC impacts parental depressive symptomatology, which can ultimately impact children is exciting to consider. There are also important clinical ramifications of this study, such as whether parental LOC can be altered using behavioral or psychological interventions to more positively impact children during their diagnosis and treatment. However, there are numerous grammatical errors that make the authors' argument difficult to follow. Additionally, correlation does not equal causation, and thus, results may need to be tempered.

We run also regression models not only correlational designs and with these statistical analyses the causation is allowed. However, we tempered a bit the results as you suggested us.

Overall Strengths

Presents data about parental LOC, which there is limited research

Depression and LOC have extensive literature, but how it relates within the cancer population is limited, so this study fills in a needed gap

There are likely clinical applications that can be extended from the current study, such as interventions geared towards shaping positive parental LOC

General Weakness

The 3 hypotheses are difficult to understand, partly due to grammatical errors and inconsistencies, as well as overall research design.

Multiple grammatical errors throughout the paper that make is difficult to read

We revised the hypotheses and a native English speaker revised the grammar and the editing.

Inconsistent use of acronyms and terminology. For example, PLOC is introduced in line 44, but sporadically utilized. Health-related LOC and parental LOC and external/internal LOC are used interchangeably, which make it difficult to follow.

We decided to use a unique terminology: PHLOC: parental health locus of control and we revised it throughout the main text. We used the term HLOC for health locus of control only when we referred to the general concept non only adopted by parents, but for example for adult cancer patients. Thank you

Inconsistent capitalization (e.g., Children vs. children)

We used only children without capitalization.

Inconsistent spelling (e.g., haematology vs. hematology)

We corrected it. Thank you

Depressive symptomology and PTSS are used almost as predictive of one another; however, these may be different psychological constructs

We specified better PTSS and depression along other psychological disorders in parents of children with leukemia. We specified better these constructs and we didn’t stress the prediction effect but only the possibility of coexist. See lines 93-98

Correlations regarding parental LOC and depressive symptomatology may not be an accurate method of answering the research questions

We firstly adopted the correlations to have a starting point to set up the regression models that is a correct method to answer the research question. We clarified better the research questions

Base rates for AML and ALL are significantly different, which is reflected in this study sample. However, given the different numbers of subjects in each cell, authors may not be able to accurately compare the 2 groups

The statistical model does not control for various factors, such as treatment intensity, length of hospital stay, etc.

We add also the treatment intensity considering the band to which belong each child with leukemia: standard, medium or high risk. We run new analysis that we add in the results section. See lines 348-354. However, we decided to maintain also the type of leukaemia, because we think it could be interesting to have this result and also because epidemiologically AML are less frequent than ALL.

Introduction

42-43: the assumption is that parental locus of control impacts parenting behaviors; however, the authors could have expanded this idea further as it seems rather abrupt. For example, does parental LOC impact parenting behaviors? And if so, how does it change parental behaviors?

Lines 38-43. We explained better these points

53-57: confusing argument, difficult to follow as the authors use examples of internal and external LOC. Furthermore, uncertain as to what authors are attempting to lay out, that external and internal LOC occur interchangeably and have both negative/positive outcomes? Or that a particular external LOC, specifically trust in other medical professionals is the only beneficial parental LOC related to treatment outcomes

Lines 63-68 were revised and the change of the place of the argument to sustain this theory (lines 68-75) clarified better these concepts.

63-66: Reference needed; however, authors write about internal LOC having poor outcomes for children’s weight problems. This argument seems misplaced and may be better served in the previous paragraph.

We changed the place of this argument, thank you

73-80: it may be beneficial to expand on what is PTSS. The end of this paragraph and the start of the next appear to make it seem that the authors are utilizing PTSS and depressive symptoms as synonymous

We clarified better these constructs. See lines 91-96

Hypotheses

Authors first introduce the concept of Current Life Perceptions, but this seems sudden; authors may need to further define this concept and how it relates to depressive symptomatology

We explained better this concept in the introduction and we reformulated the hypothesis.

102: low scores in parents’ Current Life Perceptions, is this low score good or bad?

We inserted this explanation. See lines 138-139.

104-105: high incidence of depression could be “predictive” for PTSS; however, the correlational design may not be able to answer this hypothesis accurately. (Correlation does not equal causation).

The terminology “predictive” is changed with “moderator”. We referred to the path model in the study of Tremolada et al., 2012. See line 141 and also above at lines 94-95

109-114: 2nd hypothesis is that parents will show higher scores on both external and internal LOC? I’m assuming that the authors are trying to present hypotheses based on the subdomains of parental LOC in their questionnaire; however, they do not explain or provide information about the different domains of LOC in the introduction

We cited the few studies related to this topic, even if they referred also in general to health locus of control, not strictly related to parents. We clarified this point. Lines 146-151.

115-122: very difficult to understand the 3rd hypothesis

We rewrite it and we add another hypothesis to explain better. See lines 152-155

Methods

126: Name of clinic changes compared to line 18

We changed that in the abstract.

130: reference to support authors SES stratification by “Italian norms”

We changed this point. We assessed the perceived economic condition. See lines 170-171.

133-136: authors controlled for potential differences in mother and father responses. However, mother outnumber father (91 to 13), what statistical methods were utilized to control for these differences? The authors also do not present other psychosocial information, such as family structure (i.e., single parent, extended family involvement, LGBTQ parents)

We inserted some of your comments in the limits section. See lines 235-237 about the possible parental gender differences

173: BSI-18 screens for depression, somatization, and anxiety, but given the acute nature of the situation (~1 month of cancer diagnosis), what is this measure capturing? Disease distress, psychosocial stress, or psychological mood symptoms?

We specified it. See lines 282-283

Results

189-190: what do the numbers 0.22% and 0.88% refer to? Additionally, these numbers do not add up to 1

Sorry for the mistake, we recalculated it and explained the meaning. See lines 290-293

183 -190: information can be better presented in a table

We put these results in a table. See table 1

191: first time authors introduce the terms “internal or external adoption” as it relates to locus of control. Please provide background

We erased this part from the title

198-201: difficult to comprehend what authors are presenting. What do the authors mean when they write that they “divided” parents into internal and external categories? And then they ran a paired-samples t test to compare responses of which group?

We explained better this division. See lines 305-310

204-206: hypothesis 3 changes and lists that the authors will examine SES factors, which they had not presented before in the intro, at the 2nd month, which previously did not have a time frame

High correlations between parental current and future life perceptions with BSI were noted. How did the authors address the high correlation between these variables in their regression model?

The hypothesis 3 is changed and another 4 is added. See lines 154-158. We examined the demographic (parents and children) and illness factors. We showed a possible explanation of the significative association. See lines 351-354. We put ladder of life as possible independent variable to screen depressive symptoms in parents. For this reason we put it in the regression model.

Discussion

Repeats sentences found in the abstract and intro

We checked it and we changed the sentences.

Verbiage becomes too colloquial (i.e., line 247 “huge”)

We revised it adopting the English editing services

No discussion of limit to the study

We inserted the limits and the strengths of the study. See pages 479-487.

Reviewer 3 Report

Paper by Doctor Tremolada et al. treated life perceptions and depression-associated factors in parents of children with leukemia. I consider that with the following reasons, the manuscript may be short of description for academic publication. I would like to request the authors to re-analyse and re-construct the manuscript and to try to publish it again. I consider that because the topic may contribute to the medical science, I would like to request it. 

Reasons

  1. Abstract lacks methodology description for primary outcome and analysis. Results and conclusion in Abstract may be vague. 
  2. Introduction section may be lengthy for the readers. 
  3. Methods section need to be re-written for easy understandings. 
  4.  Tables may need to be more informative for the readers. Tables need to contribute to patients, patient families or clinicians.

Author Response

Comments and Suggestions for Authors

Paper by Doctor Tremolada et al. treated life perceptions and depression-associated factors in parents of children with leukemia. I consider that with the following reasons, the manuscript may be short of description for academic publication. I would like to request the authors to re-analyse and re-construct the manuscript and to try to publish it again. I consider that because the topic may contribute to the medical science, I would like to request it.

Reasons

1. Abstract lacks methodology description for primary outcome and analysis. Results and conclusion in Abstract may be vague.

We changed the abstract along your suggestions

2. Introduction section may be lengthy for the readers.

We tried to divide it in structured sections but we decided to maintain the studies because help the reader to understand the importance of health locus of control and of the assessing of ladder of life and depression in parents.

3. Methods section need to be re-written for easy understandings.

We make some minor changes in this section

4. Tables may need to be more informative for the readers. Tables need to contribute to patients, patient families or clinicians.

Tables explained the statistical analyses. We add more comments in the conclusion that could be useful to patients, their families and clinicians (see lines 484-487) and 496-501 lines.

Round 2

Reviewer 2 Report

Overall Strengths

The authors are to be commended for their extensive work in addressing the significant concerns raised by this reviewer. Again, the data can have important clinical ramifications for supporting families during treatment for childhood cancers.

Overall Weaknesses

Continued grammatical errors throughout the text, which significantly hinders comprehension.

Abstract

Line 30: What do the authors mean by stating that "Improving…parental influence locus of control perceptions could be a preventive program to cope with parental depression symptomatology?" How does one "improve" parental influence of locus of control?

Introduction

Line 72: What do the authors mean when they say "educative…behaviors?"

Line 114: The authors address previous comments raised regarding various psychological constructs/disorders. However, what do the authors mean by "adaptation and anxiety ones?" Are these also disorders? What is an adaptation disorder?

Line 115: The authors address previous comments regarding current and future life perception, and note that these can be a "valid tool" to use as a screener. However, the authors fail to expand on the idea of what current and future life perceptions mean.

Lines 118-163 (partly addressed in line 176): The authors provide information regarding health locus of control. However, all the research that the authors cite are about an individual’s perception of HLC. For this current study, the authors are studying parental health locus of control not the child’s HLC. Is there literature regarding PHLOC impacting a child’s or parent’s health? If so, then please review in the introduction. If not, the authors can make a more sound argument.

Line 171: The authors utilize the term "moderator factor," but what does that mean? Are the authors proposing that depressive symptomatology is a moderator for PTSS? Moderating variables are typically interactions in a statistical model; however, the authors are just presenting descriptive data about levels of depressive symptomatology, not presenting a model that require moderating variables. The term "moderator" then seems misleading.

Line 180: When the authors write "children’s influence more important…" are they describing external HLC?

Line 183: "What about the possible associations with demographic and illness factors?" This seems oddly phrased; perhaps rhetorical. Consider, "what are the possible associations…" or "are there possible associations…".

Methods

Minor grammar errors.

Line 316: The authors could again add information about how the scoring on the Ladder of Life questionnaire is interpreted (i.e., low scores = worse perceptions).

Results

Line 390: The authors then further divide the LOC styles into external and internal groups. How and why do the authors divide these into this grouping? Is this how the PHLOC questionnaire was created? Additionally, "child influence" is considered an internal LOC? However, this is the parental locus of control, so if it is the child’s influence, would that not be considered an external LOC for the parent? Lastly, the authors argue above that external and internal LOC is likely on a spectrum, rather than distinct and separate groups. Therefore, are the authors not contradicting their argument by creating these dichotomous groupings?

Discussion and Conclusion

Continued grammar errors.

No further comments.

Author Response

Overall Weaknesses

Continued grammatical errors throughout the text, which significantly hinders comprehension.

We used the MDPI English proof-reading service. We attached the certification.

Abstract

Line 30: What do the authors mean by stating that "Improving…parental influence locus of control perceptions could be a preventive program to cope with parental depression symptomatology?" How does one "improve" parental influence of locus of control?

We adjusted the terminology in this sentence. Thank you

Introduction

Line 72: What do the authors mean when they say "educative…behaviors?"

We changed this term. See line 48

Line 114: The authors address previous comments raised regarding various psychological constructs/disorders. However, what do the authors mean by "adaptation and anxiety ones?" Are these also disorders? What is an adaptation disorder?

We erased this part because we didn’t analyse these disturbances in our paper. See line 90

Line 115: The authors address previous comments regarding current and future life perception, and note that these can be a "valid tool" to use as a screener. However, the authors fail to expand on the idea of what current and future life perceptions mean.

Line 91-92. We added some information. You could find other information in the literature cited and the methods section when the instruments were described in detail.

Lines 118-163 (partly addressed in line 176): The authors provide information regarding health locus of control. However, all the research that the authors cite are about an individual’s perception of HLC. For this current study, the authors are studying parental health locus of control not the child’s HLC. Is there literature regarding PHLOC impacting a child’s or parent’s health? If so, then please review in the introduction. If not, the authors can make a more sound argument.

We clarified why we decided to introduce these types of studies, even if referred to a more generic issue. We also stated throughout the paper that the literature on PHLOC was limited, so it is useful to search for other collateral research that could be indicative. See lines 95-98

Line 171: The authors utilize the term "moderator factor," but what does that mean? Are the authors proposing that depressive symptomatology is a moderator for PTSS? Moderating variables are typically interactions in a statistical model; however, the authors are just presenting descriptive data about levels of depressive symptomatology, not presenting a model that require moderating variables. The term "moderator" then seems misleading.

We changed the term with “associated”. See line 125

Line 180: When the authors write "children’s influence more important…" are they describing external HLC?

We specified it at lines 134-135. The distribution of internal and external parental health locus of control styles was then been shown in the table 2

Line 183: "What about the possible associations with demographic and illness factors?" This seems oddly phrased; perhaps rhetorical. Consider, "what are the possible associations…" or "are there possible associations…".

We changed it. See line 137. Thank you for your suggestion

Methods

Minor grammar errors.

Line 316: The authors could again add information about how the scoring on the Ladder of Life questionnaire is interpreted (i.e., low scores = worse perceptions).

We specified it. See lines 179 and 183

Results

Line 390: The authors then further divide the LOC styles into external and internal groups. How and why do the authors divide these into this grouping? Is this how the PHLOC questionnaire was created? Additionally, "child influence" is considered an internal LOC? However, this is the parental locus of control, so if it is the child’s influence, would that not be considered an external LOC for the parent? Lastly, the authors argue above that external and internal LOC is likely on a spectrum, rather than distinct and separate groups. Therefore, are the authors not contradicting their argument by creating these dichotomous groupings?

We explained better this division. See lines 328-330.

Discussion and Conclusion

Continued grammar errors.

No further comments.

Reviewer 3 Report

Thank you for improving the manuscript. I have no more concern. I appreciate the authors' efforts to report the important results. 

Author Response

Thank you for your valuable work and appreciation for our manuscript